# Seismotectonics of Shallow-Focus Earthquakes in Venezuela with Links to Gravity Anomalies and Geologic Heterogeneity Mapped by a GMT Scripting Language

**Polina Lemenkova ***🆔 **and Olivier Debeir** 🆔

Laboratory of Image Synthesis and Analysis (LISA), École Polytechnique de Bruxelles (Brussels Faculty of Engineering), Université Libre de Bruxelles (ULB), Building L, Campus de Solbosch, ULB—LISA CP165/57, Avenue Franklin D. Roosevelt 50, 1000 Brussels, Belgium
* Correspondence: polina.lemenkova@ulb.be; Tel.: +32-471860459

**Abstract:** This paper presents a cartographic framework based on algorithms of GMT codes for mapping seismically active areas in Venezuela. The data included raster grids from GEBCO, EGM-2008, and vector geological layers from the USGS. The data were iteratively processed in the console of GMT, converted by GDAL, formatted, and mapped for geophysical data visualisation; the QGIS was applied for geological mapping. We analyzed 2000 samples of the earthquake events obtained from the IRIS seismic database with a 25-year time span (1997–2021) in order to map the seismicity. The approach to linking geological, topographic, and geophysical data using GMT scripts aimed to map correlations among the geophysical phenomena, tectonic processes, geological setting, seismicity, and earthquakes. The practical application of the GMT scripts consists in automated mapping for the visualization of geological risks and hazards in the mountainous region of the Venezuelan Andes. The proposed method integrates the approach of GMT scripts with state-of-the-art GIS techniques, which demonstrated its effectiveness as a tool for mapping spatial datasets and rapid data processing in an iterative regime. In this context, using GMT and GIS to find similarities between the regional earthquake distribution and the geological and topographic setting is essential for hazard risk assessment. This study can serve as a basis for predictive seismic analysis in geologically vulnerable regions of Venezuela. In addition to a technical demonstration of GMT algorithms, this study also contributes to geological and geophysical mapping and seismic hazard assessments in South America. We present the full scripts used for mapping in a GitHub repository.

**Keywords:** risk; hazard; sustainability; earthquake; seismicity; cartography; GMT; geophysics

**PACS:** 91.10.Da; 91.10.-v; 91.10.Jf; 91.10.Op; 91.30.Dk; 93.85.Pq; 91.70.-c

**MSC:** 86Axx; 86A04; 86A15; 86A30; 86A60; 86-XX; 86-04; 86-08

**JEL Classification:** Y91; Q00; Q01; Q2; Q20; Q24; Q3; Q35; Q5; Q50; Q51; Q54; Q55; Q56; C6; C61; C63

## 1. Introduction

### 1.1. Background and Motivation

Nowadays, geological hazards are acknowledged as some of the most disastrous and high-risk events in mountainous regions. These involve a series of negative consequences for nature (landslides, tsunami, rock falls, land mass movements) and complex social problems (destruction of houses and infrastructure, injuries and losses of human lives) caused by seismic risks [1–3]. Early prediction of seismic risks in geologically vulnerable regions has become more valuable in a large variety of time-critical geospatial applications focused on evaluating geological hazards for the mitigation of earthquake hazards at the regional scale. The integration of geological and geophysical data is an effective approach for risk assessment

in mountainous regions, for the prevention of damages and danger, and for serving for early warning, long-term monitoring, management, and mitigation of seismic hazards.

Recently, advances in cartographic development and mapping have resulted in the realization of an important application of spatial data analysis: the prediction and prognosis of seismic activities or imminent earthquake events from events observed in the past [4–6]. With enough data on earthquake events, these methods enable an accurate prognosis of seismicity with the aim of mitigating hazard risks [7,8]. Early detection of seismic risks can be solved by evaluating and visualizing different geophysical determinants that affect disaster fatalities at the country level, namely, seismic hazard strength, population density, distribution of locations, demographic and socioeconomic data, ands geological background layers [9–12]. For instance, a systematic examination of the correlations and links among various geophysical and geological determinants underlying disasters is useful for the prevention of socioeconomic risks in geologically unstable zones [13,14]. In earthquake analysis, the capability of predicting earthquake events, their magnitude, and their focal depth is highly desirable for real-time risk assessment [15,16].

To build effective geological modeling approaches for seismic hazard prediction, real-time mapping applications are beneficial for earthquake prognoses. However, obtaining an automated approach to cartographic dataset processing for seismic prediction is usually difficult, resulting in biases toward a particular GIS software. The approach presented in this study aims to solve this problem by introducing an integrative study oof scripts developed in the Generic Mapping Tools (GMT) scripting toolset [17], in addition to the traditional mapping. This framework, which is different from the conventional state-of-the-art GIS methods, presents a rapid workflow of mapping based on the data processing from the console using command line and codes written in the GMT syntax. Although advanced methods of geophysical mapping of such seismically active regions as South America present a very important task, the scripting of cartography remains a new topic. To the best of our knowledge, the majority of similar studies used the conventional GIS rather than scripts for the visualization of geophysical data [18].

While conventional methods of GIS enable the plotting the maps in a traditional regime, this often requires manual operations and is not suitable for automated mapping. In such a way, it limits the types of cartographic activities that can be used for operative geological analyses and seismic prediction. In addition, data handling is limited by the restricted compatibility of formats accepted in GIS. In contrast, scripts present a rapid and repeatable procedure of automated mapping that can be integrated into real-time risk assessment and hazard analysis. As a response to these needs, GMT presents a more flexible cartographic approach and versatile strategy of spatial data processing, and it is aimed at seismic risk assessment, geological exploration, and management by using multi-source datasets.

### 1.2. Contemporaneity and Objectives

Integrated mapping using various multi-source data has the objective of visualizing a complex interplay among the geological, topographic, tectonic, and geophysical parameters. Regional effects of these settings cause the Caribbean region and Venezuela to be exposed to a high risk of earthquakes, volcanos, and associated tsunami hazards. For instance, the tectonic structures of the deep basins and erosional troughs affect the submarine geomorphology, which can be interpreted using geophysical and gravimetric measurements. Earthquake-related aftershocks, hazards, and landslides result in damage to the geotechnical infrastructure and bridges [19], demolished buildings, loss of human lives [20], and aseismic slips and creeps [21]. Such consequences affect both nature and the social sectors of Venezuela and require regular monitoring and mapping for risk and vulnerability assessment.

The visualization of the information on these processes is important for seismic prognosis and earthquake modeling. Recent research demonstrated that geological and geophysical mapping creates a basis for natural hazard risk assessment [22–26]. Therefore,

the data derived from hazard risk mapping can be used as background information and for the recognition of complex geological actions and processes. Previous cartographic investigations were mostly carried out by using traditional GIS in geological and environmental studies [27–30]. However, none of them focused on an integrated approach that would combine various cartographic techniques for the geophysical mapping of Venezuela.

Scripting techniques applied to geological mapping have not been adequately presented in the existing literature in comparison with the traditional GIS. At the same time, scripts employed for visualization of complex geophysical and geological phenomena enable automation and increase the precision of the processing of cartographic data. The use of machine-based methods enables one to accurately map and highlight correlations between geological and geophysical phenomena. With this aim, GMT scripts in combination with QGIS and its plugins were applied for the geophysical mapping of Venezuela in order to visualize the regional seismicity in connection with the geological structure and geophysical setting.

The cartographic objective of this study is to demonstrate the functionality of GMT scripts in combination with GIS techniques. The geological aim is to highlight the correlations between the geological setting, topography, geophysical anomalies, and regional seismicity through the analysis of the distribution, magnitude, and depth of earthquakes. To this end, this study applied GMT cartographic scripting techniques and QGIS in order to process high-resolution open-source raster grids and vector data. We used a variety of datasets to build the project with observed geological and geophysical processes and structures. The primary context of this work is the analysis of seismic events by using the visualization of earthquake locations in the region.

## 2. Study Area

This study focuses on Venezuela, one the of the most seismically active regions of South America. The spatial extent of the study area was 59.5°–74° W, 0°–12.5° N (Figure 1).

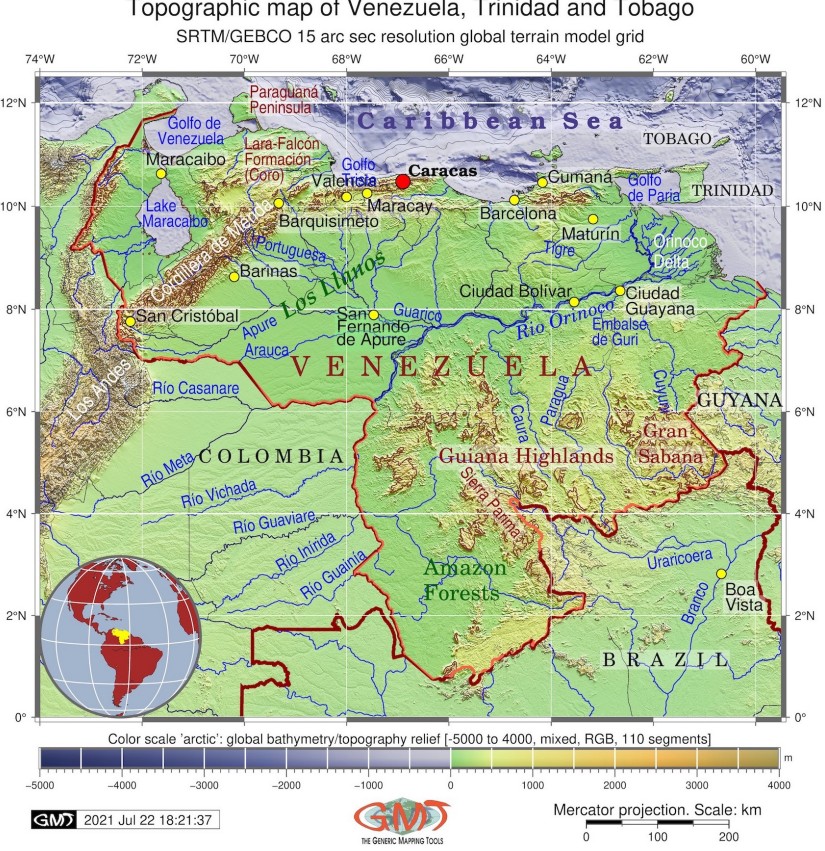

**Figure 1.** Topographic map of the region of Venezuela. Mapping: GMT. Source: authors.

Its location in the Andes exposes Venezuela to a high risk of seismic hazards. Geologically, the most important factors that have a significant influence on the seismicity in Venezuela and its surroundings include lithospheric plate subduction and the associated complex geological processes of the oblique collision of the Caribbean arc and a zone of active deformation in the eastern offshore Trinidad area [31]. This is reflected in the formation of the Venezuelan Andes as a N50°E-oriented mountain belt extending from the Colombian border in the SW to the Caribbean Sea in the NE with the associated Boconó and Valera thrust faults [32]. Seismotectonic activities in Venezuela can be characterized by a spatio-temporal composition of the constituent geophysical and geological forces, their evolutional processes, and their complex interactions. Such linkages are related to the actual location and distribution of seismic sources of the earthquakes in the Andean region with regard to the tectonics and overall dynamics of the Earth's crust in South America.

The high frequency and magnitude of earthquakes in Venezuela are associated with complex geological and geophysical settings; Figure 2. In addition, the tectonic interactions of the lithospheric plates trigger seismic activity and tectonic-related hazards that affect the lives of people and infrastructures [33–35]. The Caribbean–South American plate boundary is a very wide tectonically active zone [36,37] with recorded seismicity at various depths. Thus, anomalous series of earthquakes took place on the east coast of Trinidad with a reported depth of the main shock of 53 km [38]. A seismicity deeper than 40 km was reported on the southern end of the Lesser Antilles subduction zone and the active island arc [39].

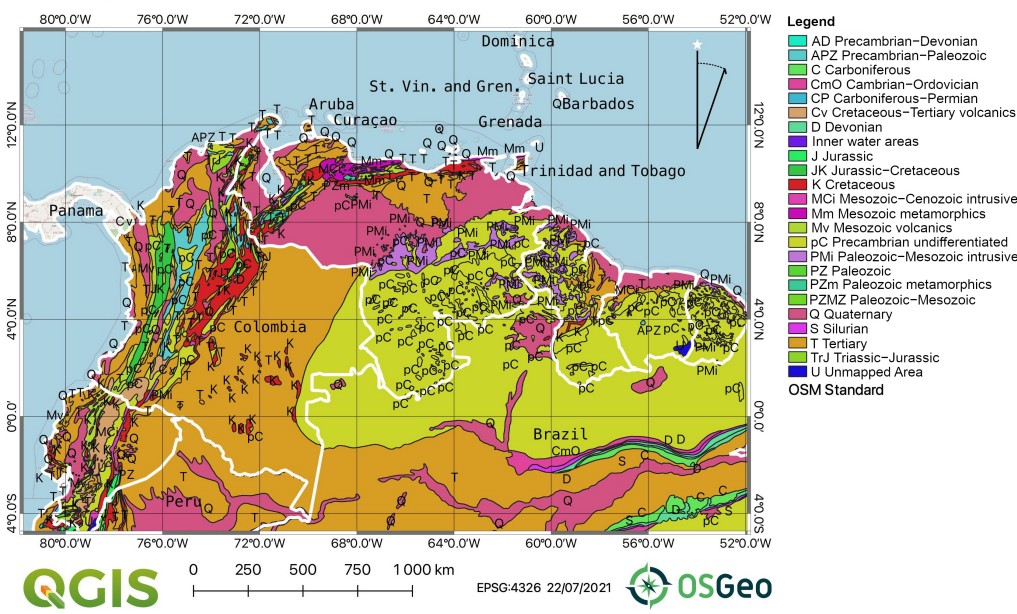

**Figure 2.** Geological map of the region of Venezuela. Mapping: QGIS. Source: authors.

Venezuela is one of the most countries that is most vulnerable to earthquakes in the Amazon Basin. The geological hazards in this region are exacerbated by social vulnerability due to the anthropogenic exposure, with a frequently injured and affected population living in the high mountainous regions of Andes, which increases the population who is at risk during earthquakes. Furthermore, catastrophic earthquakes produce a chain effect that triggers tsunamis, which amplifies the damage to the environment and human society. For example, it is reported that since the 19th century, tsunamis in the Caribbean Sea have resulted in the loss of several thousands of lives and the exposure of many people to risk and damage from landslides [40].

Venezuela is located in the north of the South American continent, where a collision of the South American and Caribbean tectonic plates causes complex regional geological patterns (Figure 2). The geodynamics of the interacting plates result in the formation of

thrust belts and foreland basins in northern Venezuela that are shaped by the interplay of the geological layers. Outcrops of Precambrian (pC), Quaternary (Q), Triassic (T), and Paleozoic–Mesozoic intrusives (PMi) dominate in the central regions of the country, while regional outcrops of Cretaceous (K) and Paleozoic metamorphics (PZm) and Mesozoic–Cenozoic intrusives (MCi) are found in the Falcón Basin (Figure 2).

The northwestern and northeastern regions of Venezuela are prone to earthquakes [41,42] due to the tectonic activity of the lithospheric plates [43,44]. According to the Incorporated Research Institutions for Seismology (IRIS) database, the repetitive occurrence of seismic events, such as volcanism and earthquakes, in northwestern Venezuela and Trinidad and Tobago was recorded as a series of events over the last 25 years. The ongoing effects of earthquakes in Venezuela [45–50] and coastal regions of Trinidad and Tobago [51,52] have made geophysical mapping an important task for seismic hazard mapping, earthquake prognosis, and risk assessment [53].

The region of the Caribbean Sea is notable for its complex geologic setting (Figure 3), where several tectonic plates experience collision, interaction, and subduction—these are the Caribbean, Cocos, Nazca, and South American plates. This results in a high seismic activity in the region. The influence of tectonic processes is reflected in a variety of topographic forms, geological structures, and geomorphic settings of Venezuela, as discussed earlier [54–57]. The geodynamic heterogeneity of the upper mantle has resulted in a specific mountain geomorphology with notable features, such as the extent of the Tobago Trough, the Lara–Falcón Basin, and the crustal structure of the Cordillera de Mérida in southwestern Venezuela, which are important segments of the Andean orogeny [58]. The distribution of earthquakes in this region is strongly controlled by the tectonic and geophysical setting, i.e., closeness to the margins of the colliding South American and Caribbean tectonic plates. Other important factors include the mountain geomorphology and topographic variability, which are shaped by the geophysical–geological forces that affect the variations in depth and magnitude of seismic events.

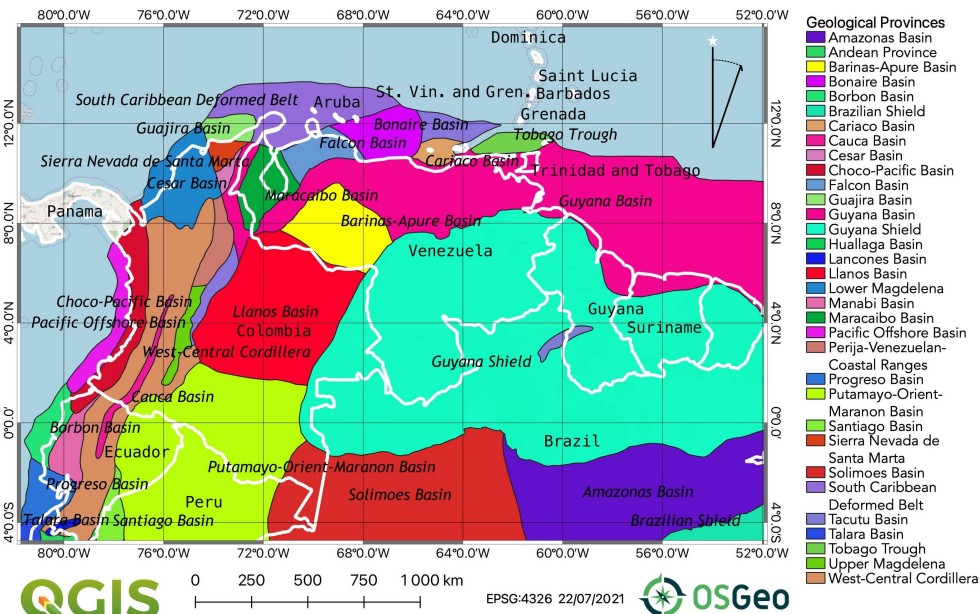

**Figure 3.** Geological provinces in the region of Venezuela. Mapping: QGIS. Source: authors.

Active tectonic movements cause the formation of orogenic mountain belts and topographies. In addition, regional bathymetry is complicated by the deep-sea trenches and troughs in the margins of the Pacific Ocean and the Caribbean Sea along the subduction path [59,60]. The complexity of the geological history is especially notable in NW

Venezuela—in the Falcón Basin—where interactions between the lithospheric plates and subducting Caribbean slab cause associated seismic activity and geological processes, such as an outcrop of igneous intrusive bodies, crustal thinning, decreased Moho depth, gravity anomalies, and rifting [61]. Previous studies [62] identified crustal thinning beneath the Falcón Basin along the western extension of the Oca–Ancón fault system, which was interpreted as a back-arc basin, as well as suture zones between the Proterozoic and Paleozoic provinces (Ouachita–Marathon-related suture) and Paleozoic and Meso-Cenozoic terranes (peri-Caribbean suture), where the seismic velocity changes.

## 3. Methodology

### 3.1. Data

Topographic, geophysical, seismic, and geological data were taken from open sources in order to present a multi-disciplinary cartographic analysis of Venezuela. The topographic data were collected from the open-access repository of the General Bathymetric Chart of the Oceans (GEBCO) in NetCDF, and they were collected in raster format with grid cells [63]. The GEBCO presents the most comprehensive data on the topography and bathymetry of the Earth with a 15 arc second resolution [64]. GEBCO data are widely used in the geosciences due to their robustness, i.e., their high resolution, precision, and reliability as a cartographic data source [65,66].

Geological data (Figures 2 and 3) were collected from the United States Geological Survey (USGS). The geological maps were processed as .shp vector layers by using the open-source QGIS software due to its compatibility with the ArcGIS data format. The visualization of these maps followed the traditional mapping process of QGIS [67]. In the first stage, vector layers of the geological provinces and outcrops were visualized within a graphical user interface (GUI). In the second stage, the study area was designated by using an overlay of the Digital Chart of the World (DCW), which is a comprehensive 1:1,000,000 scale vector base map uploaded into the QGIS project.

The geoid map was visualized based on data collected from EGM2008 [68]. The free-air gravity maps and vertical gradient were based on open-source datasets available from the public repositories of the Scripps Institution of Oceanography and described in [69]. The seismic dataset used to plot maps of earthquakes was obtained from the Incorporated Research Institutions for Seismology (IRIS) database by using the spatial extent of Venezuela and the neighboring areas of Trinidad, Tobago, and the surroundings. This dataset covered the earthquake events in the region of Venezuela for the period from 1997 to 2021 and included earthquakes with magnitudes from 1.9 to 7.3 $M_L$ on the Richter scale.

### 3.2. Methods

Maps showing the topographic, seismic, and geophysical setting of Venezuela (Figures 1 and 4–7) were made by using the scripting techniques of GMT [17] by following the existing methodology described by [70,71]. The methodological workflow included five algorithms in GMT codes, which are presented in Appendix A of this article. Prior to data visualization and mapping, the scripts were written using GMT syntax and organized in the Xcode environment. As mentioned in the previous subsection, we processed the high-resolution data by using GMT. Conventionally, the language of GMT was used in several modules for computing and processing raster data in order to plot cartographic elements. The area of interest was first selected and cut from the GEBCO grid by using the following snippet of code: $gmtgrdcutGEBCO\_2019.nc - R286/300.5/0/12.5 - Gve\_relief.nc$.

After the first step, all other substantial cartographic elements were added into a script by using the following GMT modules: psbasemap, grdimage, psscale, pstext, psclip, and grdcontour. Specifically, these included images that were visualized in selected color palettes, grids, clipped insertions of regions into a global map, coastlines, borders, rivers, bar legends on the maps, scale bars, text annotations, time stamps, etc. The technique of scripting was applied by using existing detailed descriptions [72]. For example, the 'grdcontour' module was used to compute, model, and visualize topographic isolines by

using quantitative DEM data as follows: '*gmtgrdcontourve_relief* 1.*nc* − *R* − *J* − *C*1000 − *Wthinner, darkbrown* − *O* − *K* >> $ps'. The purpose of the geoid visualization (Figure 4) in relation to the geophysical differentiation was to investigate how the gravity values responded to the effects of topographic elevations and bathymetric depressions.

While GEBCO data were gathered in NetCDF format (*GEBCO_2019.nc*), the geoid data first needed to be converted from the .adf format. This was done with the following code of GMT: '*gmtgrdconvertn*00*w*90/*w*001001.*adf geoid*_02.*grd*'. The extent and range of the data were examined through the Geospatial Data Abstraction Library (GDAL) and interpreted for the color palette as follows: '*gmtmakecpt* − *C*33_*blue_red.cpt* − *T* − 55/25 > *colors.cpt*'. The geoid was modeled by using the following code: '*gmtgrdimagegeoid*_02.*grd* − *Ccolors.cpt* − *R*286/300.5/0/12.5 − *JM*6.5*i* − *P* − *Xc* − *I* + *a*15 + *ne*0.75 − *K* > $ps'. The visualized data are shown in Figure 4.

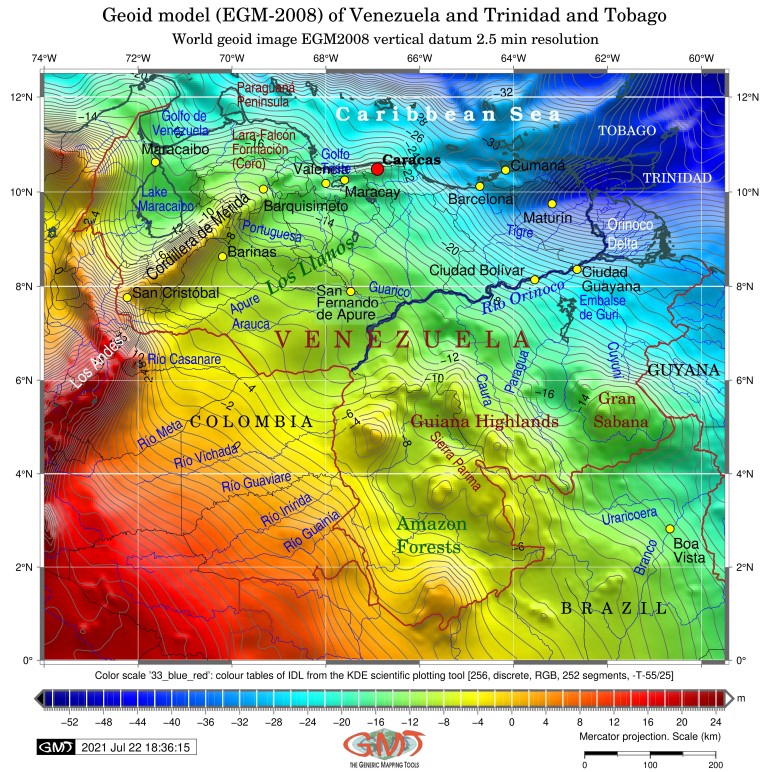

**Figure 4.** Geoid model of the region of Venezuela. Mapping: GMT. Source: authors.

The gravity grids and the geophysical data (Figures 5 and 6) were also converted from the initial raw IMG format into the GRD format, which is compatible with GMT. The code was executed as follows: '*gmtimg*2*grdgrav*_27.1.*img* − *R*286/300.5/0/12.5 − *Ggrav_VE.grd* − *T*1 − *I*1 − *E* − *S*0.1 − *V*' (for Figure 5). The same procedure was repeated for Figure 6 (vertical gradient of gravity) by following the examples described in existing work [73]. Annotations were added to all of the maps by using the 'pstext' module of GMT, as in the following example: '*gmtpstext* − *R* − *J* − *N* − *O* − *K* − *F* + *f*10*p*, 0, *black* + *jLB* + *a* − 0 >> $ps << *EOF*294.208.24*CiudadBolvarEOF*'. The data were converted into the 'ngdc' format, visualized with the 'psxy' module, and mapped, as shown in Figure 5.

The earthquakes in Figure 7 (map of seismicity) were plotted by using the 'psxy' module of GMT with the following code: '*gmtpsxy* − *R* − *Jquakes_VE.ngdc* − *Wfaint* − *i*4, 3, 6, 6*s*0.05 − *h*3 − *Scc* − *Csteps.cpt* − *O* − *K* >> $ps'. Here, '*i*4,3,6,6*s*0.05' indicates the numbers of the columns in the table (*quakes_VE.ngdc*) that were converted from the original CSV from IRIS. The volcanoes were visualized with the following code: '*gmtpsxy* − *R* − *Jvolcanoes.gmt* − *St*0.4*c* − *Gred* − *Wthinnest* − *O* − *K* >> $ps'. The complex legend showing the magnitude of events was plotted by using the following code: '*gmtpslegend* −

$R - J - Dx1.5/ -3.0 + w17.8c + o - 2.0/0.1c - F+pthin+ithinner+gwhite$ -O -K « FIN » $ps
H 10 Helvetica Seismicity: earthquakes magnitude (M) from 1.9 to 7.3 N 9 S 0.3c c 0.3c red 0.01c
0.5c M (7.1-7.3) <...> FIN'.

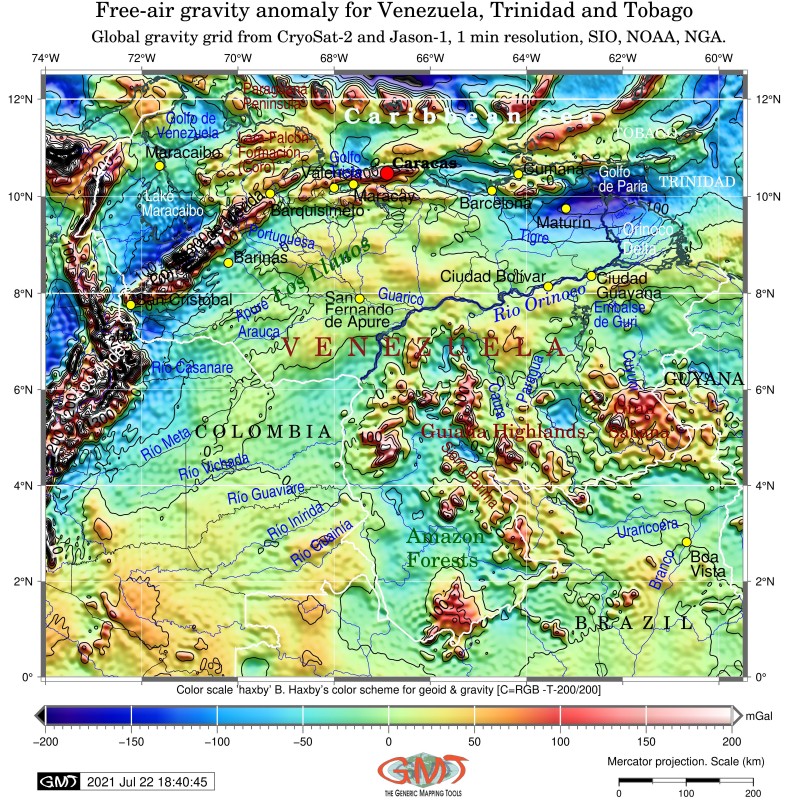

**Figure 5.** Free-air gravity model of the region of Venezuela. Mapping: GMT. Source: authors.

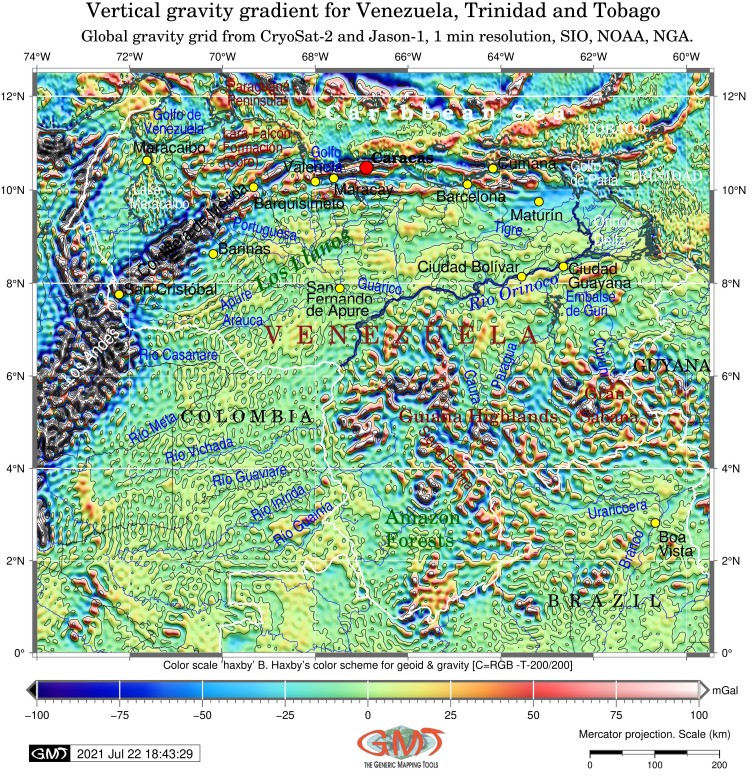

**Figure 6.** Vertical gradient of gravity in the region of Venezuela. Mapping: GMT. Source: authors.

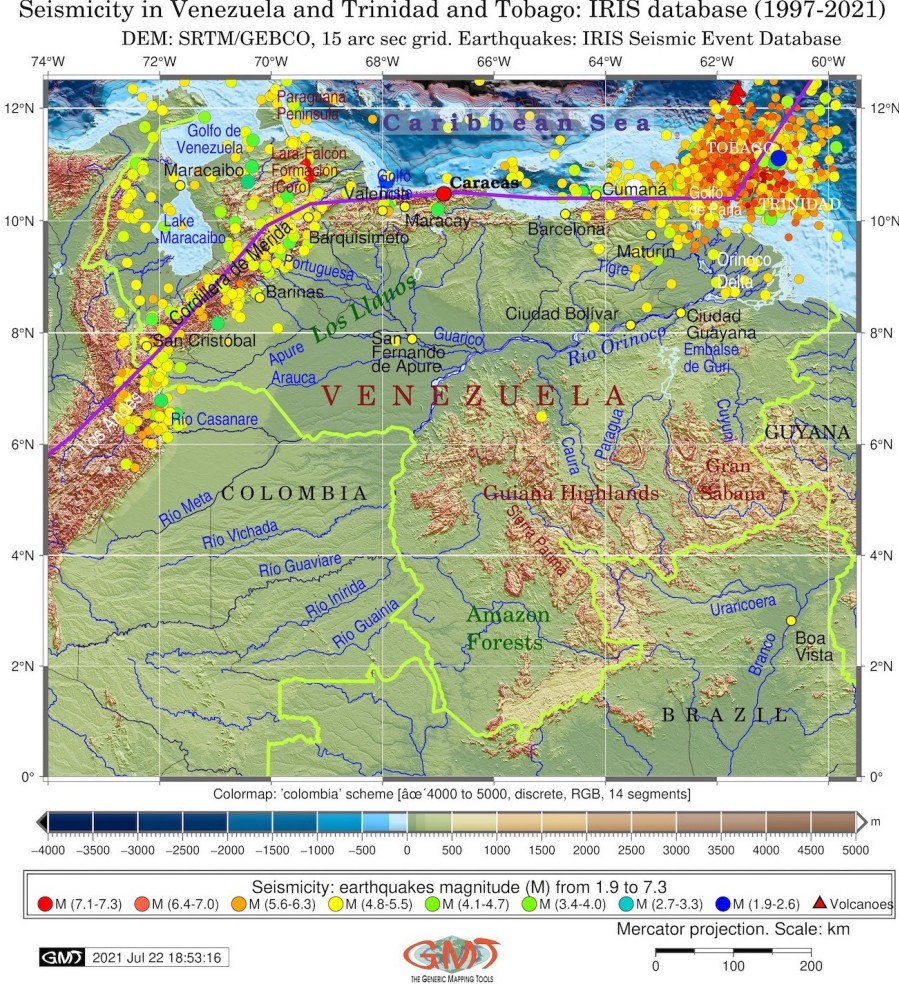

**Figure 7.** Seismic map of earthquakes in the region of Venezuela. Mapping: GMT. Source: authors.

In practice, we fused the data from seismic events to the topographic grid to show the locations of the events on the map with their attributes, which showed magnitudes ranging from 1.9 to 7.3. In our maps, we demonstrated that the density of the earthquake events visibly increased along the lines of the boundaries of lithospheric plates (thick purple line on the map in Figure 7), with a maximal concentration of events in the Andes (Cordillera de Merida) and the region of Trinidad and Tobago.

The full algorithms of the GMT codes are available in the authors' GitHub repository: https://github.com/paulinelemenkova/Mapping_Venezuela_GMT_Scripts, accessed on 1 November 2022.

## 4. Results

Figure 1 was plotted by using the GEBCO raster grid, which was selected due to its quality and reliability. A precise topographic map is essential to a fundamental understanding of the geophysics and tectonics of a region. Topographic maps enable the analysis of earthquake locations and the evaluation of the associated effects on ocean and terrestrial geomorphology. Therefore, the analysis of topography is necessary for the mapping of seismicity, risk assessment, and tsunami wave propagation. Here, the topographic and bathymetric elevations of the study area encompassing Venezuela and the islands of Trinidad and Tobago ranged from −4948 to 5536 m according to the GEBCO grid.

Figures 2 and 3 indicate the spatial extent of the geological provinces and outcrops in Venezuela. The map in Figure 2 shows that most of the territory of Venezuela is covered by the Guyana Shield (aquamarine color) in the southeastern parts of the country. The Barinas–Apure sedimentary basin is located in Llanos and the region of the Andean foothills in

western Venezuela [74]. The Barinas–Apure basin (yellow color on the map) includes oil fields, which comprise the third most important area of production in Venezuela after the Maracaibo and Eastern Venezuela basins according to the petroleum accumulation. The Falcón sedimentary basin, which produces indigenous Miocene oil from structural and stratigraphic accumulations [75], is colored in slate blue on the map.

The South Caribbean deformed belt (purple color in Figure 2) represents a submarine prism formed between the subducting tectonic plates in the Colombian and Venezuelan basins and the arc terranes along the northern rim of South America [76]. Its southwestern part, the Sinu fold belt, forms an accretionary wedge in the area of subduction of the Caribbean Plate under the South American Plate [77]. Other important geological provinces include the Guyana Basin, which stretches along the passive margin of northeastern South America, and the Maracaibo Basin (green, Figure 2). The Cariaco Basin (beige color in Figure 2) is an East–West-stretching basin [78] situated in the Gulf of Cariaco, which is located on the continental shelf of the Caribbean Sea, off the eastern coast of the surroundings of Barcelona, Figure 1).

The Tobago Trough (chartreuse green color in Figure 2) is a modern marine forearc basin of the Lesser Antilles arc and is located above the sedimentary succession. It is at least 10 km thick and is a probable oceanic basement that was formed by sediments from various stratigraphic sequences—Quaternary, early Pleistocene–late Miocene, Miocene, and late Eocene. In addition, it may also include mid-Cretaceous or older intrusive suites [79]. The distribution of the Quaternary sediments covering most of the northwest of the country (dark pink color in Figure 3) corresponds with previous studies on Plio-Quaternary extension in the Venezuelan Andes. Thus, this shows the extensional structures corresponding to elongated tilted blocks, with the geometry and kinematics of the structures corresponding to the earlier syn-orogenic extension [80].

Figure 4 shows the variations in the geoid heights over the study area. The highest values for the geoid (up to 25 m, bright red colors) are visible in the southwestern region of the map at the border with Colombia. However, the majority of the territory of Venezuela covers the extent from −50 to 8 m, with the highest values being in the mountains, showing a clearly visible correlation with topographic elevations. The undulations of the geoid in the Guyana Highlands correspond to the higher topographic elevations on the shield, with table-like 'tepuis' mountains. In general, the marine areas of the Caribbean Sea have lower geoid heights (blue-colored areas in Figure 4), which correspond well with the bathymetric depressions reflected in the geophysical setting of the Earth.

The comparison of the geoid mode in Figure 4 with the topographic and geological maps in Figures 1 and 2 shows the elevated values of the geoid that are notable in the region of the Guyana Shield. The Guyana Shield is a Precambrian geological formation on the NE coast of South America and is an area of special importance in the historical geology of Venezuela due to the high potential for mineral exploration [81]. Aside from the geological features, the environmental importance of this unique region consists in the highest biodiversity in the world, as it includes the numerous endemic species [82] of the tropical forests of Amazonia [83].

Figure 5 shows the variations in the gravity over the study area. The notable minimum of the free-air gravity coincides with distribution of the Guyana Basin, the southeastern part of Venezuela, and Trinidad and Tobago. In particular, this points to the tectonic basement depression formed during the Jurassic rifting, as well as Lake Maracaibo, an important geological region of Venezuela. The northeastern coasts of Lake Maracaibo are occupied by the Bolivar Coastal Field, the largest oil field in South America. With a basin area of 50,000 km$^2$ and oil production of $30 \times 109$ bbl , it is the second most prolific hydrocarbon basin in the world after the Middle Eastern hydrocarbon basins [84].

Maracaibo Lake has a wide variety of oil reservoir rocks with structural traps in the tectonic units, e.g., normal or inverted faults on the South American plate [85]. The oil formed in the Maracaibo Basin is mostly heavy black oil with asphaltene deposition [86,87]. The gravity in Lake Maracaibo reaches below −200 mGal in correlation with the bathymetric

depression (Figure 1) and the geological extent of the Maracaibo Basin (Figure 2). The highest values for gravity reach +200 mGal and correspond well with the topographic elevation of the Cordillera de Mérida, Columbian Andes, Guiana Highlands, and Sierra Parima. The Amazon and Orinoco Flow Basins mostly have gentle fluctuations in gravity, with values in the range between −25 and 30 mGal (Figure 5).

Figure 6 refers to a vertical gradient of normal gravity as an approximation in which a gravity measurement point, which is almost never placed on the reference ellipsoid's surface, is adjusted according to geophysical corrections [88,89]. The details regarding the distribution of vertical gravity values over Venezuela, Trinidad, and Tobago showed that the lowest values (below −100 mGal) mostly correspond to the extreme amplitude of topographic changes, e.g., the abrupt borders between the mountains/hills and the margins of the bathymetric depression, while the highest values (over 100 mGal) were detected on the tops of the mountains, which corresponded well with the local geomorphology of the region.

The distribution of the earthquakes and the seismicity of Venezuela, Trinidad, and Tobago (Figure 7) depend on the proximity of the area to a border of lithospheric plates with active margins, such as the South American and Caribbean plates, which are depicted in the map with purple thick lines. The second factor includes active geological processes (faults) and the geomorphology. Hence, the Cordillera de Mérida had the majority of the detected earthquakes after the active zone of Trinidad and Tobago. Most of the earthquakes mapped in Venezuela belonged to the 'shallow' category, that is, they had a depth between 0 and 70 km. Earthquakes at depths from 74 to 155 km were located offshore of Sucre, Venezuela, and those that were detected in Trinidad and Tobago were predominantly shallow, not exceeding 70 km. The deepest earthquakes in the inspected database were located in northern Columbia (182–200 km).

## 5. Conclusions

In this paper, we proposed two algorithms for the geophysical mapping of Venezuela, which included GMT-based scripts and traditional plotting by using QGIS, with the aim of analyzing the geohazard risk of Venezuela. Investigations and risk assessments of geohazards require advanced methods of mapping that are based on the programming of algorithms, as demonstrated in this paper. The operative use of correctly and effectively plotted maps, along with tabular data, supports reasonable and substantiated recommendations aimed at the prevention and mitigation of earthquake hazard risks in seismically active regions, such as the South American Andes. The algorithms based on GMT enabled improved performance of the cartographic workflow in comparison with that of the conventional approaches.

We demonstrated the technical advantages of the approaches to programming in cartography, which included scripting, as this is a rapidly developing branch of geoinformation. Due to the high degree of automation, scripting is the best technical cartographic solution for seismic mapping, as it enables rapid machine-based processing of data, which is beneficial for prognosis, predictive mapping, risk assessment, and hazard visualization in a real-time regime. Automation enables the rapid processing of large datasets and the effective use of multi-source heterogeneous data. Aside from that, the known advantages of scripting in cartography include machine-based graphics, which result in an aesthetically refined plotting by GMT.

Scripting cartographic methods for GMT were addressed to evaluate the geophysical and geological correspondences in a seismically active region of Venezuela and the Caribbean Sea basin. We used high-resolution geophysical, topographic, and geological datasets, such as GEBCO, EGM-2008 for geoid, gravity grids, and seismic data obtained from IRIS. The results corresponded with and supported those of previous studies on the geological and geophysical methods of gravity measurements, seismic observations, and earthquake monitoring in Venezuela and the surroundings [90,91].

This study presented a machine-based mapping workflow that contributed to the regional studies of the eastern Caribbean and Venezuela. The GMT code snippets have been presented in a GitHub repository as a demonstration of the technical possibilities of this toolset. The results showed optimized and improved cartographic performance and demonstrated the advantages of the automating processing of geophysical datasets. Multi-source geophysical, geologic, seismic, and topographic data were used for a comparative analysis of the geophysical processes. The tectonic evolution of the Caribbean Sea basin largely affected its geological setting, which could be tracked in the relevant geophysical data, indicating a high seismicity and high earthquake risk in the Andes.

Since the eastern Caribbean is a tectonically complex area that includes a subduction zone, island arc, and active volcanoes [92], a regional analysis and visualization of the varied settings were performed. Specifically, high seismicity was detected in the northwestern region of Venezuela and the northeastern region of Trinidad and Tobago, which is consistent with the geological structure associated with the tectonic setting and the related movements of the subduction of the Caribbean Plate. Although continued research is needed in order to improve earthquake monitoring in the region of Venezuela, this study presents a contribution to the seismic study of the Caribbean and South America, as it shows the earthquakes that were detected according to the IRIS database based on a 25-year time span (1997–2021).

A better understanding of the geological and tectonic factors affecting seismicity and regulating the appearance, magnitude, and focal depth of earthquakes is critical for seismically active regions, such as the Caribbean and Venezuela. Practical applications of geological risk assessment in South America include the use of information by public administrations in city management and urban planning for environmental management and the mitigation and management of hazard risk in regions prone to geological hazards, such as Venezuela.

The main contributions that we have made and the contemporaneity of the presented study are highlighted as follows:

1. Region: The region of Venezuela is at high risk of seismicity and earthquakes because it is located in the zone of the collision of the South American and Caribbean tectonic plates and the Andean orogeny. This requires spatial analysis of the regions that are at risk.
2. Geohazard: The risk assessment of geophysical hazards was based on the effective integration and visualization of multi-source data, which provided detailed insights into the environmental and geological settings of Venezuela and supported complex investigations in seismically active regions of South America.
3. Data: Geological disasters are related to high seismicity, exposure to hazards, and vulnerability of people at risk. This requires complex and detailed studies that summarize these factors and visualize regional geophysical settings, as shown in this paper.
4. Methods: The combination of scripting for GMT and GIS is a solid foundation for cartographic data analysis. An efficient method for processing multi-format data ensures hazard mapping and geophysical and geological visualization with the aim of preventing and evaluating seismic hazards.

In this paper, we approached the problem of automated mapping in the geophysical sciences and the estimation of the level of seismicity in the region of Venezuela for risk assessment studies as a contribution from the new perspective of data analysis. The scripting approach was mainly designed for the integration of data and the presented maps for studies of risk assessment, though it is also valuable as a general-purpose mapping technique. The limitations of GMT may be constrained in specific tasks, such as the analysis of remote sensing data or image processing, where general-purpose languages, such as Python and R, perform better.

The methodology that was presented and explained here for plotting maps with codes from a console can be potentially extended to other regions of the world, since the approach to data analysis remains identical. It is possible to reuse the presented codes for other

seismically active regions of the world. The only modifications of the method would include the spatial extent of the study area, that is, the cartographic coordinates for plotting the maps and for downloading the data from the IRIS catalogue.

The implementation of the GMT scripts for other regions implies that one only needs to access datasets for a given region and to adjust the codes to this regional extent in order to map the data fairly well. We consider this cartographic scripting approach as a promising extension of our method for similar future work on seismicity and risk assessment. We demonstrated that the performance and visualization that were obtained are effective and that the level of workflow automation is high. The experimental results verify the usefulness of our approach for both of the following research objectives: the evaluation of the cartographic performance of GMT and the mapping of seismicity in the region of Venezuela and the northern Andes.

**Author Contributions:** Supervision, conceptualization, methodology, software, resources, funding acquisition, and project administration, O.D.; writing—original draft preparation, methodology, software, data curation, visualization, formal analysis, validation, writing—review and editing, and investigation, P.L. All authors have read and agreed to the published version of the manuscript.

**Funding:** This project was supported by the Federal Public Planning Service Science Policy or Belgian Science Policy Office, Federal Science Policy—BELSPO (B2/202/P2/SEISMOSTORM).

**Institutional Review Board Statement:** Not applicable.

**Informed Consent Statement:** Not applicable.

**Data Availability Statement:** The GitHub repository contains the GMT scripts used for the mapping in this study: https://github.com/paulinelemenkova/Mapping_Venezuela_GMT_Scripts, accessed on 1 November 2022.

**Acknowledgments:** The authors thank the three anonymous reviewers for their careful reading, critical suggestions, and useful comments that helped improve an earlier version of this manuscript.

**Conflicts of Interest:** The authors declare no conflict of interest.

## Abbreviations

The following abbreviations are used in this manuscript:

| | |
|---|---|
| CSV | Comma-separated values |
| DCW | Digital Chart of the World |
| EGM | Earth Gravitational Models |
| GEBCO | General Bathymetric Chart of the Oceans |
| GIS | Geographic Information System |
| GMT | Generic Mapping Tools |
| GUI | Graphical User Interface |
| QGIS | Quantum GIS |
| IRIS | Incorporated Research Institutions for Seismology |
| NetCDF | Network Common Data Form |
| NGDC | National Geophysical Data Center |
| USGS | United States Geological Survey |

## Appendix A. GMT Scripts

*Appendix A.1. GMT Script for Topographic Mapping*

**Listing A1:** GMT code used to plot Figure 1 (topographic map).

```
1 #!/bin/sh
2 # Purpose: shaded relief grid raster map from the GEBCO dataset (here: Venezuela)
3 # GMT modules: gmtset, gmtdefaults, grdcut, makecpt, grdimage, psscale, grdcontour,
       psbasemap, gmtlogo, psconvert
4 # http://soliton.vm.bytemark.co.uk/pub/cpt-city/arendal/tn/arctic.png.index.html
5 # GMT set up
6 gmt set FORMAT_GEO_MAP=dddF \
7 # Overwrite defaults of GMT
8 gmtdefaults -D > .gmtdefaults
9 #chsh -s /bin/bash
```

```
10  chsh -s /bin/zsh
11  gmt grdcut GEBCO_2019.nc -R286/300.5/0/12.5 -Gve_relief.nc
12  gmt grdcut ETOPO1_Ice_g_gmt4.grd -R286/300.5/0/12.5 -Gve_relief1.nc
13  gdalinfo ve_relief.nc -stats
14  # Minimum=-4947.963, Maximum=5535.625, Mean=162.220, StdDev=788.097
15  # create mask of vector layer from the DCW of country's polygon
16  gmt pscoast -R286/300.5/0/12.5 -Dh -M -EVE > ve.txt
17  # Make color palette
18  gmt makecpt -Carctic.cpt > pauline.cpt
19  # Generate a file
20  ps=Topography_VE.ps
21  # Make background transparent image
22  gmt grdimage ve_relief.nc -Cpauline.cpt -R286/300.5/0/12.5 -JM6i -P -I+a15+ne0.75 -t20 -
        Xc -K > $ps
23  # Add isolines
24  gmt grdcontour ve_relief1.nc -R -J -C500 -W0.1p -O -K >> $ps
25  # Add coastlines, borders, rivers
26  gmt pscoast -R -J -P \
27      -Ia/thinner,blue -Na -N1/thickest,darkred -W0.1p -Df -O -K >> $ps
28  # CLIPPING
29  # 1. Start: clip the map by mask to only include country
30  gmt psclip -R286/300.5/0/12.5 -JM6.0i ve.txt -O -K >> $ps
31  # 2. create map within mask
32  # Add raster image
33  gmt grdimage ve_relief.nc -Cpauline.cpt -R286/300.5/0/12.5 -JM6.0i -I+a15+ne0.75 -Xc -P
        -O -K >> $ps
34  # Add isolines
35  gmt grdcontour ve_relief1.nc -R -J -C1000 -Wthinner,darkbrown -O -K >> $ps
36  # Add coastlines, borders, rivers
37  gmt pscoast -R -J \
38      -Ia/thinner,blue -Na -N1/thicker,tomato -W0.1p -Df -O -K >> $ps
39  # 3: Undo the clipping
40  gmt psclip -C -O -K >> $ps
41  # Add color barlegend
42  gmt psscale -Dg286/-1.0+w15.2c/0.4c+h+o0.0/0i+ml -R -J -Cpauline.cpt \
43      -Bg500a1000f100+l"Color scale 'arctic': global bathymetry/topography relief [-5000
        to 4000, mixed, RGB, 110 segments]" \
44      -I0.2 -By+lm -O -K >> $ps
45  # Add grid
46  gmt psbasemap -R -J \
47      --MAP_FRAME_AXES=WEsN \
48      --FORMAT_GEO_MAP=ddd:mm:ssF \
49      -Bpx4f2a2 -Bpyg4f2a2 -Bsxg4 -Bsyg2 \
50      -B+t"Topographic map of Venezuela, Trinidad and Tobago" -O -K >> $ps
51  # Add scalebar, directional rose
52  gmt psbasemap -R -J \
53      -Lx12.7c/-2.2c+c50+w200k+l"Mercator projection. Scale: km"+f \
54      -UBL/-5p/-65p -O -K >> $ps
55  # Texts
56  gmt pstext -R -J -N -O -K \
57  -F+jTL+f10p,21,darkred+jLB+a-0 -Gwhite@60 >> $ps << EOF
58  290.1 11.8 Peninsula
59  EOF
60  # insert map
61  # Countries codes: ISO 3166-1 alpha-2. Continent codes AF (Africa), AN (Antarctica), AS
        (Asia), EU (Europe), OC (Oceania), NA (North America), or~SA (South America). -EEU+
        ggrey
62  gmt psbasemap -R -J -O -K -DjBL+w3.3c+stmp >> $ps
63  read x0 y0 w h < tmp
64  gmt pscoast --MAP_GRID_PEN_PRIMARY=thinner,white -Rg -JG293/6N/$w -Da -Gbrown -A5000 -Bg
        -Wfaint -ESA+gpeachpuff -EVE+gyellow -Slightsteelblue3 -O -K -X$x0 -Y$y0 >> $ps
65  #gmt pscoast -Rg -JG12/5N/$w -Da -Gbrown -A5000 -Bg -Wfaint -ECM+gbisque -O -K -X$x0 -
        Y$y0 >> $ps
66  gmt psxy -R -J -O -K -T  -X-${x0} -Y-${y0} >> $ps
67  # Add GMT logo
68  gmt logo -Dx6.2/-2.9+o0.1i/0.1i+w2c -O -K >> $ps
69  # Add subtitle
70  gmt pstext -R0/10/0/15 -JX10/10 -X0.5c -Y5.0c -N -O \
71      -F+f10p,Helvetica,black+jLB >> $ps << EOF
72  2.3 13.3 SRTM/GEBCO 15 arc sec resolution global terrain model grid
73  EOF
74  # Convert to image file using GhostScript
75  gmt psconvert Topography_VE.ps -A0.5c -E720 -Tj -Z
```

*Appendix A.2. GMT Script for Geoid Mapping*

**Listing A2:** GMT code used to plot Figure 4 (geoid modeling).

```
1  #!/bin/sh
2  # Purpose: geoid of Venezuela
```

```
 3  # GMT modules: gmtset, gmtdefaults, grdcut, makecpt, grdimage, psscale, grdcontour,
        psbasemap, gmtlogo, psconvert
 4  # http://soliton.vm.bytemark.co.uk/pub/cpt-city/kst/tn/33_blue_red.png.index.html
 5  # GMT set up
 6  gmt set FORMAT_GEO_MAP=dddF \
 7  # Overwrite defaults of GMT
 8  gmtdefaults -D > .gmtdefaults
 9  gmt grdconvert n00w90/w001001.adf geoid_02.grd
10  gdalinfo geoid_02.grd -stats
11  # Minimum=-70.856, Maximum=32.958, Mean=-24.768, StdDev=19.081
12  # Generate a color palette table from grid
13  gmt makecpt -C33_blue_red.cpt -T-55/25 > colors.cpt
14  # Generate a file
15  ps=Geoid_VE.ps
16  gmt grdimage geoid_02.grd -Ccolors.cpt -R286/300.5/0/12.5 -JM6.5i -P -Xc -I+a15+ne0.75 -
        K > $ps
17  # Add shorelines
18  gmt grdcontour geoid_02.grd -R -J -C1.0 -A2.0+f8p,0,black -Wthinner,dimgray -O -K >> $ps
19  # Add grid
20  gmt psbasemap -R -J \
21      --MAP_FRAME_AXES=WEsN \
22      --FORMAT_GEO_MAP=ddd:mm:ssF \
23      -Bpx4f2a2 -Bpyg4f2a2 -Bsxg4 -Bsyg2 \
24      --MAP_TITLE_OFFSET=0.8c \
25      --FONT_ANNOT_PRIMARY=7p,0,black \
26      --FONT_LABEL=7p,25,black \
27      --FONT_TITLE=13p,25,black \
28      -B+t"Geoid model (EGM-2008) of Venezuela and Trinidad and Tobago" -O -K >> $ps
29  # Add legend
30  gmt psscale -Dg286/-1.0+w16.5c/0.4c+h+o0.0/0i+ml+e -R -J -Ccolors.cpt \
31      -Bg2f0.2a4+l"Color scale '33_blue_red': colour tables of IDL from the KDE scientific
            plotting tool [256, discrete, RGB, 252 segments, -T-55/25]" \
32      -I0.2 -By+lm -O -K >> $ps
33  # Add scale, directional rose
34  gmt psbasemap -R -J \
35      --FONT=7p,0,black \
36      --FONT_ANNOT_PRIMARY=6p,0,black \
37      --MAP_TITLE_OFFSET=0.1c \
38      --MAP_ANNOT_OFFSET=0.1c \
39      -Lx14.7c/-2.3c+c50+w200k+l"Mercator projection. Scale (km)"+f \
40      -UBL/-5p/-65p -O -K >> $ps
41  # Add coastlines, borders, rivers
42  gmt pscoast -R -J -P -Ia/thinnest,blue -Na -N1/thick,brown -Wthick,darkslategray -Df -O
        -K >> $ps
43  # Texts
44  gmt pstext -R -J -N -O -K \
45  -F+jTL+f10p,21,darkred+jLB+a-0 >> $ps << EOF
46  290.0 10.5 (Coro)
47  EOF
48  # Add GMT logo
49  gmt logo -Dx7.0/-2.9+o0.1i/0.1i+w2c -O -K >> $ps
50  # Add subtitle
51  gmt pstext -R0/10/0/15 -JX10/10 -X0.1c -Y5.9c -N -O \
52      -F+f10p,25,black+jLB >> $ps << EOF
53  3.0 13.6 World geoid image EGM2008 vertical datum 2.5 min resolution
54  EOF
55  # Convert to image file using GhostScript
56  gmt psconvert Geoid_VE.ps -A1.0c -E720 -Tj -Z
```

*Appendix A.3. GMT Script for Free-Air Gravity Mapping*

**Listing A3:** GMT code used to plot Figure 5 (free-air gravity modeling).

```
 1  #!/bin/sh
 2  # Purpose: mapping free-air gravity anomaly of Venezuela
 3  # GMT modules: gmtset, gmtdefaults, img2grd, makecpt, grdimage, psscale, grdcontour,
        psbasemap, gmtlogo, psconvert, pscoast
 4  # http://soliton.vm.bytemark.co.uk/pub/cpt-city/pj/4/index.html
 5  # GMT set up
 6  gmt set FORMAT_GEO_MAP=dddF \
 7      MAP_FRAME_PEN=dimgray \
 8  # Overwrite defaults of GMT
 9  gmtdefaults -D > .gmtdefaults
10  gmt img2grd grav_27.1.img -R286/300.5/0/12.5 -Ggrav_VE.grd -T1 -I1 -E -S0.1 -V
11  gdalinfo grav_VE.grd -stats
12  # Minimum=-218.668, Maximum=942.457, Mean=5.694, StdDev=64.118
13  # Generate a color palette table from grid
14  gmt makecpt -Chaxby -T-200/200 > colors.cpt
15  # Generate a file
16  ps=Grav_VE.ps
```

```
17  gmt grdimage grav_VE.grd -Ccolors.cpt -R286/300.5/0/12.5 -JM6.0i -P -I+a15+ne0.75 -Xc -K
        > $ps
18  # Add isolines
19  gmt grdcontour grav_VE.grd -R -J -C50 -A100 -Wthinner -O -K >> $ps
20  # Add grid
21  gmt psbasemap -R -J \
22      --MAP_FRAME_AXES=WEsN \
23      --FORMAT_GEO_MAP=ddd:mm:ssF \
24      -Bpx4f2a2 -Bpyg4f2a2 -Bsxg4 -Bsyg2 \
25      -B+t"Free-air gravity anomaly for Venezuela, Trinidad and Tobago" -O -K >> $ps
26  # Add legend
27  gmt psscale -Dg286/-1.0+w15.0c/0.4c+h+o0.0/0i+ml+e -R -J -Ccolors.cpt \
28      --FONT_LABEL=7p,Helvetica,black \
29      --FONT_ANNOT_PRIMARY=7p,0,black \
30      --FONT_TITLE=8p,25,black \
31      -Bg50f5a50+l"Color scale 'haxby' B. Haxby's color scheme for geoid & gravity [C=RGB
        -T-200/200]" \
32      -I0.2 -By+l"mGal" -O -K >> $ps
33  # Add scale, directional rose
34  gmt psbasemap -R -J \
35      -Lx14.0c/-2.3c+c50+w200k+l"Mercator projection. Scale (km)"+f \
36      -UBL/-5p/-70p -O -K >> $ps
37  # Add coastlines, borders, rivers
38  gmt pscoast -R -J -P -Ia/thinnest,blue -Na -N1/thick,white -Wthin,darkslategray -Df -O -
        K >> $ps
39  # Texts
40  gmt pstext -R -J -N -O -K \
41  -F+jTL+f14p,32,darkgreen+jLB+a-330 >> $ps << EOF
42  290.8 7.8 Los Llanos
43  EOF
44  # Add GMT logo
45  gmt logo -Dx7.0/-2.9+o0.1i/0.1i+w2c -O -K >> $ps
46  # Add subtitle
47  gmt pstext -R0/10/0/15 -JX10/10 -X0.0c -Y4.8c -N -O \
48      -F+f10p,25,black+jLB >> $ps << EOF
49  1.0 13.6 Global gravity grid from CryoSat-2 and Jason-1, 1 min resolution, SIO, NOAA,
        NGA.
50  EOF
51  # Convert to image file using GhostScript
52  gmt psconvert Grav_VE.ps -A0.5c -E720 -Tj -Z
```

## Appendix A.4. GMT Script for Vertical Gradient of Gravity

**Listing A4:** GMT code used to plot Figure 5 (free-air gravity modeling).

```
1   #!/bin/sh
2   # Purpose: geophysical mapping of Venezuela (vertical free-air gravity gradient)
3   # GMT modules: gmtset, gmtdefaults, img2grd, makecpt, grdimage, psscale, grdcontour,
        psbasemap, gmtlogo, psconvert, pscoast
4   # http://soliton.vm.bytemark.co.uk/pub/cpt-city/pj/4/index.html
5   # GMT set up
6   gmt set FORMAT_GEO_MAP=dddF \
7   # Overwrite defaults of GMT
8   gmtdefaults -D > .gmtdefaults
9   gmt img2grd curv_27.1.img -R286/300.5/0/12.5 -Ggrav_v_VE.grd -T1 -I1 -E -S0.1 -V
10  gdalinfo grav_v_VE.grd -stats
11  # Minimum=-488.036, Maximum=870.350, Mean=0.358, StdDev=43.643
12  # Generate a color palette table from grid
13  gmt makecpt -Chaxby -T-100/100 > colors.cpt
14  # Generate a file
15  ps=Grav_VE_v.ps
16  gmt grdimage grav_v_VE.grd -Ccolors.cpt -R286/300.5/0/12.5 -JM6.0i -P -I+a15+ne0.75 -Xc
        -K > $ps
17  # Add isolines
18  gmt grdcontour grav_v_VE.grd -R -J -C100 -Wthinnest -O -K >> $ps
19  # Add grid
20  gmt psbasemap -R -J \
21      --MAP_FRAME_AXES=WEsN \
22      --FORMAT_GEO_MAP=ddd:mm:ssF \
23      -Bpx4f2a2 -Bpyg4f2a2 -Bsxg4 -Bsyg2 \
24      -B+t"Vertical gravity gradient for Venezuela, Trinidad and Tobago" -O -K >> $ps
25  # Add legend
26  gmt psscale -Dg286/-1.0+w15.0c/0.4c+h+o0.0/0i+ml+e -R -J -Ccolors.cpt \
27      -Bg25f2.5a25+l"Color scale 'haxby' B. Haxby's color scheme for geoid & gravity [C=
        RGB -T-200/200]" \
28      -I0.2 -By+l"mGal" -O -K >> $ps
29  # Add scale, directional rose
30  gmt psbasemap -R -J \
31      -Lx14.0c/-2.3c+c50+w200k+l"Mercator projection. Scale (km)"+f \
32      -UBL/-5p/-70p -O -K >> $ps
```

```
33  # Add coastlines, borders, rivers
34  gmt pscoast -R -J -P -Ia/thinnest,blue -Na -N1/thick,white -Wthin,darkslategray -Df -O -
        K >> $ps
35  # Texts
36  gmt pstext -R -J -N -O -K \
37  -F+jTL+f12p,21,white+jLB+a-315 >> $ps << EOF
38  286.7 5.8 Los Andes
39  EOF
40  # Add GMT logo
41  gmt logo -Dx7.0/-2.9+o0.1i/0.1i+w2c -O -K >> $ps
42  # Add subtitle
43  gmt pstext -R0/10/0/15 -JX10/10 -X0.0c -Y4.8c -N -O \
44      -F+f10p,25,black+jLB >> $ps << EOF
45  1.0 13.6 Global gravity grid from CryoSat-2 and Jason-1, 1 min resolution, SIO, NOAA,
        NGA.
46  EOF
47  # Convert to image file using GhostScript
48  gmt psconvert Grav_VE_v.ps -A0.5c -E720 -Tj -Z
```

*Appendix A.5. GMT Script for Seismic Mapping*

**Listing A5:** GMT codes used to plot Figure 7 (seismic mapping).

```
1   #!/bin/sh
2   # Purpose: seismicity map of Venezuela (earthquake distribution according to the IRIS
        database (1997-2021) of seismic events)
3   # GMT modules: gmtset, gmtdefaults, grdcut, makecpt, grdimage, psscale, grdcontour,
        psbasemap, gmtlogo, psconvert
4   # http://soliton.vm.bytemark.co.uk/pub/cpt-city/wkp/shadowxfox/tn/colombia.png.index.
        html
5   # GMT set up
6   gmt set FORMAT_GEO_MAP=dddF
7   gmtdefaults -D > .gmtdefaults
8   # Extract a topographic subset of Venezuela and surroundings:
9   gmt grdcut ETOPO1_Ice_g_gmt4.grd -R286/300.5/0/12.5 -Gve_relief1.nc
10  gmt grdcut GEBCO_2019.nc -R286/300.5/0/12.5 -Gve_relief.nc
11  gdalinfo -stats ve_relief.nc
12  # Min=-4947.963 Max=5535.625
13  # Make color palette
14  gmt makecpt -Ccolombia.cpt > pauline.cpt
15  gmt makecpt -Cseis -T1.9/7.3/0.5 -Z > steps.cpt
16  ps=Seis_VE.ps
17  # Make raster image
18  gmt grdimage ve_relief.nc -Cpauline.cpt -R286/300.5/0/12.5 -JM6.5i -I+a15+ne0.75 -Xc -P
        -K > $ps
19  # Add legend
20  gmt psscale -Dg286/-1.0+w16.5c/0.4c+h+o0.0/0i+ml+e -R -J -Cpauline.cpt \
21      -Bg500f50a500+l"Colormap..." \
22      -I0.2 -By+lm -O -K >> $ps
23  # Add isolines
24  gmt grdcontour ve_relief1.nc -R -J -C500 -Wthinnest,brown -O -K >> $ps
25  # Add coastlines, borders, rivers
26  gmt pscoast -R -J -P \
27      -Ia/thinner,blue -Na -N1/thickest,olivedrab1 -Wthin,lightcyan2 -Df -O -K >> $ps
28  # Add grid
29  gmt psbasemap -R -J \
30      --MAP_FRAME_AXES=WEsN \
31      --FORMAT_GEO_MAP=ddd:mm:ssF \
32      -Bpx4f2a2 -Bpyg4f2a2 -Bsxg4 -Bsyg2 \
33      -B+t"Seismicity in Venezuela and Trinidad and Tobago: IRIS database (1997-2021)" -O
        -K >> $ps
34  # Add scale, directional rose
35  gmt psbasemap -R -J \
36      -Lx14.0c/-3.6c+c50+w200k+l"Mercator projection. Scale: km"+f \
37      -UBL/-5p/-110p -O -K >> $ps
38  # Add earthquake points
39  # separator in numbers of table: dot (.), not comma ! (British style)
40  gmt psxy -R -J quakes_VE.ngdc -Wfaint -i4,3,6,6s0.05 -h3 -Scc -Csteps.cpt -O -K >> $ps
41  # Add geological lines and points
42  gmt psxy -R -J volcanoes.gmt -St0.4c -Gred -Wthinnest -O -K >> $ps
43  # fabric and magnetic lineation picks fracture zones
44  gmt psxy -R -J GSFML_SF_FZ_KM.gmt -Wthicker,goldenrod1 -O -K >> $ps
45  gmt psxy -R -J GSFML_SF_FZ_RM.gmt -Wthicker,pink -O -K >> $ps
46  gmt psxy -R -J ridge.gmt -Sf0.5c/0.15c+l+t -Wthick,red -Gyellow -O -K >> $ps
47  gmt psxy -R -J ridge.gmt -Sc0.05c -Gred -Wthickest,red -O -K >> $ps
48  # tectonic plates
49  gmt psxy -R -J TP_Caribbean.txt -L -Wthickest,purple -O -K >> $ps
50  gmt psxy -R -J TP_Cocos.txt -L -Wthickest,purple -O -K >> $ps
51  gmt psxy -R -J TP_Nazca.txt -L -Wthickest,purple -O -K >> $ps
52  gmt psxy -R -J TP_South_Am.txt -L -Wthickest,purple -O -K >> $ps
```

```
53  # Texts
54  gmt pstext -R -J -N -O -K \
55  -F+f10p,17,black+jLB+a-0 >> $ps << EOF
56  293.20 10.58 Caracas
57  EOF
58  # Processed likewise for similar texts
59  gmt pslegend -R -J -Dx1.5/-3.0+w17.8c+o-2.0/0.1c \
60      -F+pthin+ithinner+gwhite \
61      --FONT=8p,black -O -K << FIN >> $ps
62  H 10 Helvetica Seismicity: earthquakes magnitude (M) from 1.9 to 7.3
63  N 9
64  S 0.3c c 0.3c red 0.01c 0.5c M (7.1-7.3)
65  S 0.3c c 0.3c tomato 0.01c 0.5c M (6.4-7.0)
66  S 0.3c c 0.3c orange 0.01c 0.5c M (5.6-6.3)
67  S 0.3c c 0.3c yellow 0.01c 0.5c M (4.8-5.5)
68  S 0.3c c 0.3c chartreuse1 0.01c 0.5c M (4.1-4.7)
69  S 0.3c c 0.3c chartreuse1 0.01c 0.5c M (3.4-4.0)
70  S 0.3c c 0.3c cyan3 0.01c 0.5c M (2.7-3.3)
71  S 0.3c c 0.3c blue 0.01c 0.5c M (1.9-2.6)
72  S 0.3c t 0.3c red 0.03c 0.5c Volcanoes
73  FIN
74  # Add GMT logo
75  gmt logo -Dx7.0/-4.5+o0.1i/0.1i+w2c -O -K >> $ps
76  # Add subtitle
77  gmt pstext -R0/10/0/15 -JX10/10 -X0.5c -Y5.9c -N -O \
78      -F+f11p,25,black+jLB >> $ps << EOF
79  1.0 13.6 DEM: SRTM/GEBCO, 15 arc sec grid. Earthquakes: IRIS Seismic Event Database
80  EOF
81  # Convert to image file using GhostScript
82  gmt psconvert Seis_VE.ps -A1.7c -E720 -Tj -Z
```

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
