# Peer review of "Seismotectonics of Shallow-Focus Earthquakes in Venezuela with Links to Gravity Anomalies and Geologic Heterogeneity Mapped by a GMT Scripting Language"

_sustainability, doi:10.3390/su142315966_

Round 1

Reviewer 1 Report

sustainability-2045749  Review. 

This study integrates GMT scripts with GIS techniques for spatial mapping in a seismotectonic study of earthquakes in Venezuela.  The study is very thorough and well explained by the authors and represents a solid contribution to this field of study, in particular in the area of geohazards.  There are some minor spelling problems in places, but nothing serious.  Or maybe they are just typographical errors.  There are small errors in English grammar and usage in a few places in the paper, but again, they are not serious, and the paper is still quite readable and understandable.  No serious scientific or technical questions came to mind as I was reading through the paper.  Therefore, I think the paper can be published in its present form. 

Author Response

Dear Editors of the Sustainability,

Please find attached the revised version of the paper. We have carefully followed the comments and suggestions of the reviewers and corrected the manuscript accordingly.

All the corrections in the text are marked up yellow for Track Changes.

The replies to the comments of the reviewers are listed below.

Using the opportunity, we thank the reviewers for careful reading of the paper which improved the initial version of the manuscript.

With kind regards, - Authors (Polina Lemenkova and Olivier Debeir).

23.11.2022.

Reviewer 1

No

Reviewer’s Comments

Author’s actions

1

This study integrates GMT scripts with GIS techniques for spatial mapping in a seismotectonic study of earthquakes in Venezuela. The study is very thorough and well explained by the authors and represents a solid contribution to this field of study, in particular in the area of geohazards.

Thank you very much for the endorsement of our manuscript. We highly appreciate your review, encouragement and positive feedback.

2

There are some minor spelling problems in places, but nothing serious. Or maybe they are just typographical errors. There are small errors in English grammar and usage in a few places in the paper, but again, they are not serious, and the paper is still quite readable and understandable.

We have proofread the manuscript. We corrected all occasional typesetting misprints and minor grammar mistakes where necessary. Spelling and punctuation are checked everywhere in the text.

3

No serious scientific or technical questions came to mind as I was reading through the paper. Therefore, I think the paper can be published in its present form.

Many thanks for supporting the publication of our paper.

Open Review

Original Review Report

English language and style

( ) English very difficult to understand/incomprehensible
( ) Extensive editing of English language and style required
( ) Moderate English changes required
(x) English language and style are fine/minor spell check required
( ) I don't feel qualified to judge about the English language and style

Yes

Can be improved

Must be improved

Not applicable

Is the content succinctly described and contextualized with respect to previous and present theoretical background and empirical research (if applicable) on the topic?

(x)

( )

( )

( )

Are all the cited references relevant to the research?

(x)

( )

( )

( )

Are the research design, questions, hypotheses and methods clearly stated?

(x)

( )

( )

( )

Are the arguments and discussion of findings coherent, balanced and compelling?

(x)

( )

( )

( )

For empirical research, are the results clearly presented?

(x)

( )

( )

( )

Is the article adequately referenced?

(x)

( )

( )

( )

Are the conclusions thoroughly supported by the results presented in the article or referenced in secondary literature?

(x)

( )

( )

( )

Comments and Suggestions for Authors

sustainability-2045749  Review. 

This study integrates GMT scripts with GIS techniques for spatial mapping in a seismotectonic study of earthquakes in Venezuela.  The study is very thorough and well explained by the authors and represents a solid contribution to this field of study, in particular in the area of geohazards.  There are some minor spelling problems in places, but nothing serious.  Or maybe they are just typographical errors.  There are small errors in English grammar and usage in a few places in the paper, but again, they are not serious, and the paper is still quite readable and understandable.  No serious scientific or technical questions came to mind as I was reading through the paper.  Therefore, I think the paper can be published in its present form. 

Submission Date

04 November 2022

Date of this review

09 Nov 2022 02:00:38

Reviewer 2 Report

I don't have any suggestions for revising this manuscript. I think this manuscript can be published in the current version

Author Response

Dear Editors of the Sustainability,

Our replies to the comments of the reviewer are listed below. Using the opportunity, we thank the reviewers for careful reading of the paper and support of the publication.

With kind regards, - Authors (Polina Lemenkova and Olivier Debeir).

23.11.2022.

Reviewer 2

No

Reviewer’s Comments

Author’s actions

1

Comments and Suggestions for Authors

I don't have any suggestions for revising this manuscript. I think this manuscript can be published in the current version

Thank you very much for supporting the publication of our paper. We highly appreciate the endorsement of our manuscript, encouragement and positive feedback. Many thanks!

Open Review

Original Review Report

English language and style

( ) English very difficult to understand/incomprehensible
( ) Extensive editing of English language and style required
( ) Moderate English changes required
(x) English language and style are fine/minor spell check required
( ) I don't feel qualified to judge about the English language and style

Yes

Can be improved

Must be improved

Not applicable

Is the content succinctly described and contextualized with respect to previous and present theoretical background and empirical research (if applicable) on the topic?

(x)

( )

( )

( )

Are all the cited references relevant to the research?

(x)

( )

( )

( )

Are the research design, questions, hypotheses and methods clearly stated?

(x)

( )

( )

( )

Are the arguments and discussion of findings coherent, balanced and compelling?

(x)

( )

( )

( )

For empirical research, are the results clearly presented?

(x)

( )

( )

( )

Is the article adequately referenced?

(x)

( )

( )

( )

Are the conclusions thoroughly supported by the results presented in the article or referenced in secondary literature?

(x)

( )

( )

( )

Comments and Suggestions for Authors

I don't have any suggestions for revising this manuscript. I think this manuscript can be published in the current version

Submission Date

04 November 2022

Date of this review

20 Nov 2022 13:24:08

Reviewer 3 Report

The paper proposes a cartographic framework based on GMT code methods for mapping seismically active regions in Venezuela. In order to depict regional seismicity in relation to the geologic structure, the GMT method of scripting cartographic toolsets in conjunction with QGIS and its plugins is used to develop geophysical maps of Venezuela. The full scripts of the framework are shared in the GitHub repository.

The topic is interesting, and the results are clearly addressed and discussed throughout the manuscript. Yet, there are some minor comments that should be addressed. I can see the value in the proposed manuscript, and my recommendation is that the paper should undergo a minor revision before publication.

Minor comments to the different sections are reported in the following:

1.     How can the authors comment on the integration of these maps for risk assessment studies? What could be the possible limitations of the study? And how could these codes be implemented in other regions of the world? There should be a discussion in the manuscript on this point.

2.     I propose to have a separate section for the Study Area. The Introduction section should only include the state of the art, novelty, and organization of the paper.

3.     I recommend removing the GMT codes from the manuscript and sharing them through a link (GitHub).

4.     In the conclusion section, there is no need to divide the section into two parts. It is recommended to be combined.

Editorial Comments:

1.     Line 658: Capitalise “the”.

2.     Lines 677-678: In the sentence “The majority of mapped earthquakes InVenezuela belong to the category ..” the word “in” should be separated.

Author Response

Dear Editors of the Sustainability,

Please find attached the revised version of the paper. We have carefully followed the comments and suggestions of the reviewers and corrected the manuscript accordingly.

All the corrections in the text are marked up yellow for Track Changes.

The replies to the comments of the reviewers are listed below.

Using the opportunity, we thank the reviewers for careful reading of the paper which improved the initial version of the manuscript.

With kind regards, - Authors (Polina Lemenkova and Olivier Debeir).

23.11.2022.

Reviewer 3

No

Reviewer’s Comments

Author’s actions

1

Are the research design, questions, hypotheses and methods clearly stated? – Can be improved.

The Methodology section is updated. We separated 2 different subsections – 3.1. Data and 3.2. Methods. In 3.1. Data we described data sources and their major characteristics. In 3.2. Methods we described all the methodological steps: scripting workflow, processing data in QGIS, etc. The proofreading of this section is done to correct some wordings and minor misprints where required.

2

Are the arguments and discussion of findings coherent, balanced and compelling? – Can be improved.

We have updated the results and Conclusions section and added some more comments and explanations regarding the presented maps. The scripts are also described with more details and added explanations. The Conclusions section is updated and proofread.

3

Comments and Suggestions for Authors

The paper proposes a cartographic framework based on GMT code methods for mapping seismically active regions in Venezuela. In order to depict regional seismicity in relation to the geologic structure, the GMT method of scripting cartographic toolsets in conjunction with QGIS and its plugins is used to develop geophysical maps of Venezuela. The full scripts of the framework are shared in the GitHub repository.

The topic is interesting, and the results are clearly addressed and discussed throughout the manuscript. Yet, there are some minor comments that should be addressed. I can see the value in the proposed manuscript, and my recommendation is that the paper should undergo a minor revision before publication. Minor comments to the different sections are reported in the following:

Thank you very much for the supporting and encouraging comments. We have updated the manuscript following all your comments. Additionally, we have proofread the manuscript throughout. We corrected all the occasional typesetting misprints and checked once more again spelling and punctuation. Minor grammar errors are corrected are checked everywhere in the text.

4

How can the authors comment on the integration of these maps for risk assessment studies?

What could be the possible limitations of the study?

And how could these codes be implemented in other regions of the world?

There should be a discussion in the manuscript on this point.

Answered and added the following discussion in the text (lines 417–437, Conclusions):

In this paper, we approached the problem of automated mapping in geophysical sciences and estimation of the seismicity level in Venezuela region for the risk assessment studies as a contribution from the new perspective of data analysis. The scripting approach is mainly designed for integration of data and the presented maps for risk assessment studies while being valuable as a general purpose mapping technique as well. The limitations of the GMT may be constrained by the specific tasks, such as remote sensing data analysis or image processing where general purpose languages, such as Python and R perform better.

The presented and explained methodology of plotting maps by codes from the console can be potentially extended for other regions of the world, since the approach to data analysis remains identical. It is possible to reuse the presented codes to other seismically active regions of the world. The only modifications of the method will include the spatial extent of the study areas, that is, cartographic coordinates for plotting the maps and for downloading the data from the IRIS catalogue.

The implementation of the GMT scripts for other regions implies that we need only to access dataset for the given region and to adjust the codes to this regional extent to map the data fairly well. We consider cartographic scripting approach as a promising extension of our method for future similar works on seismicity and risk assessment. We demonstrated that the performance and obtained visualization are effective and the workflow automation is high. The experimental results verify the usefulness of our approach for both research objectives: evaluating the cartographic performance of GMT and mapping the seismicity in Venezuela region and northern Andes”

5

I propose to have a separate section for the Study Area. The Introduction section should only include the state of the art, novelty, and organization of the paper.

Corrected, as suggested. The Introduction section now contains 2 subsections – 1.1. Background and motivation; and 1.2. Actuality and objectives.

We made a new separate section – 2. ‘Study Area’ and moved there all the paragraphs and phrases related to the geologic and tectonic structure of Venezuela, and illustrating 3 maps. Here we explained the particular features of this regions, seismic instability and briefly outlines the main steps in the tectonic development.

6

I recommend removing the GMT codes from the manuscript and sharing them through a link (GitHub).

We have moved all the codes with 5 listings from the Methodology section to the Appendix, to the end of the paper. In this way, it can be accessed by the reader, if necessary, but does not interrupt to the textual flow of the text. Also, the link to the GitHub repository is provided, additionally.

7

In the conclusion section, there is no need to divide the section into two parts. It is recommended to be combined.

Corrected, as suggested. The Conclusion section is now a single block of text, without sub-divisions. Some minor rewording are done in the text.

8

Editorial Comments:

1.     Line 658: Capitalise “the”.

2.     Lines 677-678: In the sentence “The majority of mapped earthquakes InVenezuela belong to the category..” the word “in” should be separated.

Corrected both issues.

Open Review

Original Peer Review

English language and style

( ) English very difficult to understand/incomprehensible
( ) Extensive editing of English language and style required
( ) Moderate English changes required
(x) English language and style are fine/minor spell check required
( ) I don't feel qualified to judge about the English language and style

Yes

Can be improved

Must be improved

Not applicable

Is the content succinctly described and contextualized with respect to previous and present theoretical background and empirical research (if applicable) on the topic?

(x)

( )

( )

( )

Are all the cited references relevant to the research?

(x)

( )

( )

( )

Are the research design, questions, hypotheses and methods clearly stated?

( )

(x)

( )

( )

Are the arguments and discussion of findings coherent, balanced and compelling?

( )

(x)

( )

( )

For empirical research, are the results clearly presented?

(x)

( )

( )

( )

Is the article adequately referenced?

(x)

( )

( )

( )

Are the conclusions thoroughly supported by the results presented in the article or referenced in secondary literature?

(x)

( )

( )

( )

Comments and Suggestions for Authors

The paper proposes a cartographic framework based on GMT code methods for mapping seismically active regions in Venezuela. In order to depict regional seismicity in relation to the geologic structure, the GMT method of scripting cartographic toolsets in conjunction with QGIS and its plugins is used to develop geophysical maps of Venezuela. The full scripts of the framework are shared in the GitHub repository.

The topic is interesting, and the results are clearly addressed and discussed throughout the manuscript. Yet, there are some minor comments that should be addressed. I can see the value in the proposed manuscript, and my recommendation is that the paper should undergo a minor revision before publication.

Minor comments to the different sections are reported in the following:

1.     How can the authors comment on the integration of these maps for risk assessment studies? What could be the possible limitations of the study? And how could these codes be implemented in other regions of the world? There should be a discussion in the manuscript on this point.

2.     I propose to have a separate section for the Study Area. The Introduction section should only include the state of the art, novelty, and organization of the paper.

3.     I recommend removing the GMT codes from the manuscript and sharing them through a link (GitHub).

4.     In the conclusion section, there is no need to divide the section into two parts. It is recommended to be combined.

Editorial Comments:

1.     Line 658: Capitalise “the”.

2.     Lines 677-678: In the sentence “The majority of mapped earthquakes InVenezuela belong to the category ..” the word “in” should be separated.

Submission Date

04 November 2022

Date of this review

18 Nov 2022 18:08:35
